# Unveiling Options with Neural Decomposition

**Mahdi Alikhasi and Levi H. S. Lelis**
Amii, Department of Computing Science, University of Alberta
`{alikhasi,levi.lelis}@ualberta.ca`

## Abstract

In reinforcement learning, agents often learn policies for specific tasks without the ability to generalize this knowledge to related tasks. This paper introduces an algorithm that attempts to address this limitation by decomposing neural networks encoding policies for Markov Decision Processes into reusable sub-policies, which are used to synthesize temporally extended actions, or options. We consider neural networks with piecewise linear activation functions, so that they can be mapped to an equivalent tree that is similar to oblique decision trees. Since each node in such a tree serves as a function of the input of the tree, each sub-tree is a sub-policy of the main policy. We turn each of these sub-policies into options by wrapping it with while-loops of varied number of iterations. Given the large number of options, we propose a selection mechanism based on minimizing the Levin loss for a uniform policy on these options. Empirical results in two grid-world domains where exploration can be difficult confirm that our method can identify useful options, thereby accelerating the learning process on similar but different tasks.

## 1 Introduction

A key feature of intelligent agents that learn by interacting with the environment is their ability to transfer what is learned in one task to another (André & Markovitch, 2005; Taylor & Stone, 2009). In this paper, we investigate extracting from neural networks "helpful" temporally extended actions, or options (Sutton et al., 1999), as a means of transferring knowledge across tasks. Given a neural network encoding a policy for a task, where a policy is a function that returns an action for a given state of the task, we decompose the network into sub-networks that are used to create options.

We assume that neural networks encoding policies use two-part piecewise-linear activation functions, such as ReLU (Nair & Hinton, 2010), which is standard in reinforcement learning (Mnih et al., 2013). As shown in previous work (Lee & Jaakkola, 2020; Orfanos & Lelis, 2023), such networks can be mapped into oblique decision trees (Breiman et al., 1986), which is a type of tree where each node encodes a linear function of the input. In this paper, we map networks with piecewise linear functions to a structure similar to an oblique decision tree, which we call a neural tree. Like an oblique decision tree, each internal node in the neural tree is a function of the input of the network. Unlike an oblique tree, the leaf nodes of neural trees represent the entire output layer of the network. Thus, each node in the neural tree, including leaf nodes, represents a sub-policy of the neural policy. We hypothesize that some of these sub-policies can generalize across tasks and we turn them into options by wrapping them in while-loops that perform different numbers of iterations.

Since the number of sub-policies grows exponentially with the number of neurons in the network, we introduce a procedure to select a subset of options that minimizes the Levin loss (Orseau et al., 2018) on a set of tasks that the agent has already mastered. We compute the Levin loss for the uniform policy over the options because such a policy approximates the policy encoded in a randomly initialized neural network representing an agent in the early stages of learning. If the tasks used to generate and select options are similar to the next task, minimizing the Levin loss can increase the chances that the agent visits promising states early in learning, thus guiding the agent's exploration.

Our method of learning options has benefits compared to previous approaches. First, it enables the extraction of options from neural policies, even when learning options was not the original intention. This implies the possibility of learning options from "legacy agents", provided that their networks use piecewise-linear functions. Second, our method automatically learns from data when to start

and when to terminate an option, and it also decides the number of options required to accelerate learning. The disadvantages are that it assumes a sequence of tasks in which the options generalize, and it involves a combinatorial search problem to select options that minimize the Levin loss.

We empirically evaluated on two grid-world problems, where exploration can be difficult, the hypothesis that neural decomposition can unveil helpful options. We used small neural networks to allow for the evaluation of all sub-policies; this choice prevented us from possibly conflating a lack of helpful options with our inability to find them. Compared to a baseline that generates options from the non-decomposed policy of trained networks, options learned with our decomposition were more effective in speeding up learning. Our options were also more effective than transfer learning baselines and three methods that learn options for a specific task. This paper offers a novel approach to learning options, where options are not intentionally learned, but extracted from existing policies.

The implementation used in our experiments is available online.[1]

## 2 RELATED WORK

**Options**   Temporally extended actions have a long history in reinforcement learning. Many previous works rely on human knowledge to provide the options (Sutton et al., 1999) or components to enable them to be learned, such as the duration of the options (Frans et al., 2017; Tessler et al., 2017), the number of options to be learned (Bacon et al., 2017; Igl et al., 2020), or human supervision (Andreas et al., 2017). Our decomposition-based method learns all options components from data generated by the agent interacting with the environment. Other approaches use specific neural architectures for learning options (Achiam et al., 2018), while we show that options can "occur naturally" in neural policies, even when it was not intended to learn them. Options have also been used to improve exploration in reinforcement learning (Machado et al., 2018; Jinnai et al., 2020; Machado et al., 2023). We minimize the Levin loss for the uniform policy with the goal of guiding exploration in the early stages of learning. However, instead of covering the space, we equip the agent with the ability to sample sequences of actions that led to promising states in previous tasks.

**Transfer Learning**   Transfer learning approaches are represented by different categories such as regularization-based, such as Elastic Weight Consolidation (Kirkpatrick et al., 2017), which prevent the agent from becoming too specialized in a given task. Others focused on adapting the neural architecture to allow for knowledge transfer across tasks, while retaining the agent's ability of learning new skills. Progressive Neural Networks (Rusu et al., 2016), Dynamically Expandable Networks (Yoon et al., 2017), and Progress and Compress (Schwarz et al., 2018) are representative methods of this approach. Previous work also stored past experiences as a way to allow the agent to learn new skills while retaining old ones (Rolnick et al., 2019). Previous work has also transferred the weights of one model to the next (Narvekar et al., 2020). One can transfer the weights of a policy (Clegg et al., 2017), of a value network, or both (Shao et al., 2018). SupSup (Wortsman et al., 2020) and Modulating Masks (Ben-Iwhiwhu et al., 2022) also transfer knowledge by learning masks for different tasks. The use of masks is particularly related to our approach because they also allow the agent to use sub-networks of a network, but to learn policies instead of options.

**Compositional Methods**   Compositional method attempts to decompose the problem so that one can train sub-policies for the decomposed sub-problems (Kirsch et al., 2018; Goyal et al., 2019; Mendez et al., 2022). These methods assume that the problem can be decomposed and often rely on domain-specific knowledge to perform such a decomposition. $\pi$-PRL learns sub-policies in earlier tasks, and these sub-policies are made available as actions to the agent (Qiu & Zhu, 2021). We also learn policies in earlier tasks as a means of learning options, but we consider all sub-policies of a policy to learn options, instead of considering only the final policy as an option, as in $\pi$-PRL.

We use representative baselines of these categories in our experiments, including Option-Critic (Bacon et al., 2017), ez-greedy (Dabney et al., 2021), and DCEO (Klissarov & Machado, 2023). We also use Progressive Neural Networks, and a variant of our method that does not use decomposition and therefore resembles the skill learning process used in $\pi$-PRL and H-DRLN (Tessler et al., 2017).

---

[1]`https://github.com/lelis-research/Dec-Options`

## 3 PROBLEM DEFINITION

We consider Markov decision processes (MDPs) $(S, A, p, r, \gamma)$, where $S$ represents the set of states and $A$ is the set of actions. The function $p(s_{t+1}|s_t, a_t)$ encodes the transition model, since it gives the probability of reaching state $s_{t+1}$ given that the agent is in $s_t$ and takes action $a_t$ at time step $t$. When moving from $s_t$ to $s_{t+1}$, the agent observes the reward value of $R_{t+1}$, which is returned by the function $r$; $\gamma$ in $[0, 1]$ is the discount factor. A policy $\pi$ is a function that receives a state $s$ and an action $a$ and returns the probability in which $a$ is taken in $s$. The objective is to learn a policy $\pi$ that maximizes the expected sum of discounted rewards for $\pi$ starting in $s_t$: $\mathbb{E}_{\pi,p}[\sum_{k=0}^{\infty} \gamma^k R_{k+t+1}|s_t]$.

Let $\mathcal{P} = \{\rho_1, \rho_2, \cdots, \rho_n\}$ be a set of MDPs, which we refer to as tasks, for which the agent learns to maximize the expected sum of discounted rewards. After learning policies for $\mathcal{P}$, we evaluate the agent while learning policies for a set of tasks $\mathcal{P}'$ with $\mathcal{P}' \cap \mathcal{P} = \emptyset$. This is a simplified version of scenarios where agents learn continually; we focus on transferring knowledge from $\mathcal{P}$ to $\mathcal{P}'$ through options (Konidaris & Barto, 2007).

## 4 LEARNING OPTIONS WITH NEURAL NETWORK DECOMPOSITION

We use the knowledge that the agent generates while learning policies $\pi$ for tasks in $\mathcal{P}$ to learn temporally extended actions that use "parts" of $\pi$ that can be "helpful". We hypothesize that these temporally extended actions can speed up the learning process of policies for $\mathcal{P}'$. We consider the options framework to define temporally extended actions (Sutton et al., 1999). An option $\omega$ is a tuple $(I_\omega, \pi_\omega, T_\omega)$, where $I_\omega$ is the initiation set of states in which the option can be selected; $\pi_\omega$ is the policy that the agent follows once the option starts; $T_\omega$ is a function that receives a state $s_t$ and returns the probability in which the option terminates in $s_t$. We consider the call-and-return execution of options: once $\omega$ is initiated, the agent follows $\pi_\omega$ until it terminates.

Here is an overview of our algorithm for learning options.

1. Learn a set of neural policies $\{\pi_{\theta_1}, \pi_{\theta_2}, \cdots, \pi_{\theta_n}\}$, one for each task in $\mathcal{P}$.

2. Decompose each neural network encoding $\pi_{\theta_i}$ into a set of sub-policies $U_i$ (Section 4.1).

3. Select a subset from $\cup_{i=1}^{n} U_i$ to form a set of options $\Omega$ (Section 4.2).

4. Use $A \cup \Omega$ as the set of actions that the agent has available to learn a policy for tasks in $\mathcal{P}'$.

We can use any algorithm that learns a parameterized policy $\pi_\theta$, such as policy gradient (Williams, 1992) and actor-critic algorithms (Konda & Tsitsiklis, 1999) in Step 1 above. In Step 4, we can use any algorithm to solve MDPs, because we augment the agent's action space with the options learned in Steps 1–3 (Kulkarni et al., 2016). The process of decomposing the policies into sub-policies is described in Section 4.1 (Step 2) and the process of defining and selecting a set of options is described in Section 4.2 (Step 3). Since we use the set of options $\Omega$ as part of the agent action space for the tasks in $\mathcal{P}'$, Step 3 only defines $\pi_\omega$ and $T_\omega$ for all $\omega$ in $\Omega$, and $I_\omega$ is set to be all states $S$. Due to its process of decomposing trained neural networks, we call DEC-OPTIONS both the algorithm and the options it learns.

### 4.1 DECOMPOSING NEURAL POLICIES INTO SUB-POLICIES

We consider fully connected neural networks with $m$ layers $(1, \cdots, m)$, where the first layer is given by the input values $X$ and the $m$-th layer the output of the network. For example, $m = 3$ for the network shown in Figure 1. Each layer $j$ has $n_j$ neurons $(1, \cdots, n_j)$ where $n_1 = |X|$. The parameters between layers $i$ and $i + 1$ of the network are indicated by $W^i \in \mathbb{R}^{n_{i+1} \times n_i}$ and $B^i \in \mathbb{R}^{n_{i+1} \times 1}$. The $k$-th row vector of $W^i$ and $B^i$, denoted $W_k^i$ and $B_k^i$, represent the weights and the bias term of the $k$-th neuron of the $(i + 1)$-th layer. In Figure 1, $n_1 = 2$ and $n_2 = 2$. Let $A^i \in \mathbb{R}^{n_i \times 1}$ be the values the $i$-th layer produces, where $A^1 = X$ and $A^m$ is the output of the model. A forward pass in the model computes the values of $A^i = g(Z^i)$, where $g(\cdot)$ is an activation function and $Z^i = W^{i-1} \cdot A^{i-1} + B^{i-1}$. In Figure 1, the neurons in the hidden layer use ReLU as the activation function, and the output neuron uses a Sigmoid function.

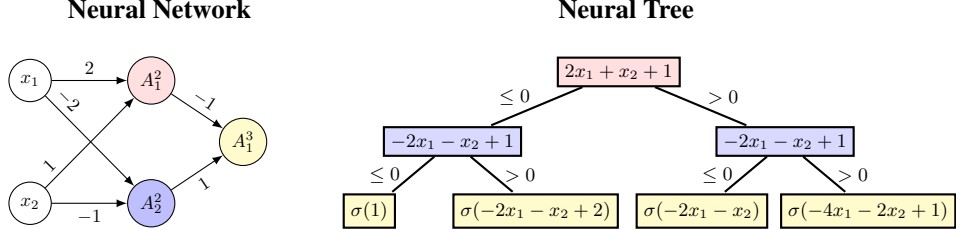

Figure 1: A neural network with two inputs, two ReLU neurons in the hidden layer, and one Sigmoid neuron in the output neuron is shown on the left. All bias terms of the model are 1; for simplicity, we omit bias values. The neural tree representing the same function encoded in the network is shown on the right. The root of the tree represents the neuron $A_1^2$, the nodes in the second layer represent the neuron $A_2^2$, and the leaf nodes represent the output neuron $A_1^3$, where $\sigma(\cdot)$ is the Sigmoid function. The colors of the neurons match the colors of the nodes in the tree that represent them.

Let $(N, E)$ be a binary tree, which we call a *neural tree*. Here, $N$ represents the nodes in the tree and $E$ the connections between them. Given a network with two-part piecewise linear activation functions (e.g., ReLU), we can construct an equivalent neural tree, where each internal node of the tree represents a neuron in the layers $[2, \cdots, m-1]$ of the network (all neurons, but those in the output layer). The neurons in the output layer are represented in each leaf node of the tree. Like oblique decision trees, each internal node of neural trees defines a function $P \cdot X + v \leq 0$ of the input $X$. Unlike oblique decision trees, each leaf node represents the computation of the output layer of the network. In the case of a neural policy for an MDP with $|A| = 2$ actions, so that the number of output neurons is $n_m = 1$, each leaf node returns the Sigmoid value of $P \cdot X + v$. If the number of actions is $|A| > 2$, so that $n_m = |A|$, then each leaf node returns the probability distribution given by the Softmax values of $P' \cdot X + V$, where $P' \in \mathbb{R}^{n_m \times n_1}$ and $V \in \mathbb{R}^{n_1 \times 1}$. In continuous action spaces, each leaf returns the parameterized distribution from which an action can be sampled. Once the parameters of the internal and leaf nodes are defined, inference is made starting at the root of the tree, and if $P \cdot X + v \leq 0$, then we follow the left child; otherwise, we follow the right child. This process is repeated until a leaf node is reached, where the leaf computation is performed.

If a neuron uses a two-part piecewise linear function $g(\cdot)$, its output value is determined by one of the two linear functions composing $g$. For example, a neuron employing a ReLU function, expressed as $g(z) = \max(0, z)$. The neuron's output is either 0 or $z$. When the input of a ReLU network is fixed, each neuron is either inactive (yielding 0) or active (yielding $z$). This leads to the concept of an activation pattern, an ordered set composed of binary values representing each node in the network. It signals whether a node is active or inactive for an input $X$ (Montúfar et al., 2014; Zhang et al., 2018; Lee & Jaakkola, 2020). Every path in a neural tree corresponds to an activation pattern, excluding those in the output layer. For example, if $P \cdot X + v \leq 0$ is true for the root of the tree, then the left sub-tree of the root represents the scenario in which the first neuron of the network is inactive. In contrast, the right sub-tree represents the scenario in which the first neuron is active. Choosing a path in the neural tree means that every neuron represents a linear function. Consequently, combining multiple linear functions results in another linear function. This allows us to define the function of the output layer in each tree as a linear function of the input $X$.

**Example 1** *Consider the example in Figure 1, where the neural network represents a neural policy with two actions, given by the Sigmoid value of the output neuron. Figure 1 also shows the neural tree that represents the neural network. The tree accounts for all activation patterns in the neural network. For example, if both neurons in the hidden layer are inactive, then $A_1^2 = A_2^2 = 0$ and the output of the network is $\sigma(0 \cdot -1 + 0 \cdot 1 + 1) = \sigma(1)$; this activation pattern is represented by the left branch of the tree. If the first neuron (from top to bottom) in the hidden layer is active and the second is inactive, the neural network produces the output $\sigma(-1 \cdot (2x_1 + x_2 + 1) + 1 \cdot 0 + 1)) = \sigma(-2x_1 - x_2)$; this activation pattern is given by following the right and then left branch from the root.*

Since all nodes in the neural tree represent a function of the input, each sub-tree of the neural tree represents a sub-policy of the policy the network encodes. A neural network with $d$ neurons and a single hidden layer decomposes into $2^{d+1} - 1$ sub-policies, one for each node in the tree. Note, however, that the order in which neurons are represented along the paths of the neural tree could

result in different sub-policies. In our running example, the sub-programs we obtain if $A_1^2$ is the root of the tree are different from the programs we obtain if $A_2^2$ is the root. The total number of different sub-policies for a network with a single hidden layer with $d$ neurons is $\sum_{i=0}^{d} \binom{d}{i} \cdot 2^i$. In our example, this sum results in $1 + 4 + 4 = 9$. To give intuition, the value of 1 for $i = 0$ represents the sub-policy that is identical to the original policy. The value of 4 for $i = 1$ is the number of sub-policies given by trees rooted at the children of the root of the tree: the root of the tree can represent 2 different neurons and each of them has 2 children, for a total of 4. Finally, the sub-policies given by the leaf nodes are identical independently of order of the earlier nodes, and they are exactly 4. In this paper, we consider small neural networks with one hidden layer, so we can evaluate all sub-policies of a neural policy. This is to test our hypothesis that these sub-policies result in "helpful" options.

## 4.2 SYNTHESIZING AND SELECTING OPTIONS

Let $\{\pi_{\theta_1}, \pi_{\theta_2}, \cdots, \pi_{\theta_n}\}$ be the set of policies that the agent learns for each task in $\mathcal{P}$. Let $U_i$ be the set of sub-policies obtained through the neural tree of $\pi_{\theta_i}$, as described in the previous section, and $U = \{U_1, U_2, \cdots, U_n\}$. Let $\{(s_0, a_0), (s_1, a_1), \cdots, (s_T, a_T)\}$ be a sequence of state-action pairs observed under $\pi$ and a distribution of initial states $\mu$ for a task $\rho$, where $s_0$ is sampled from $\mu$ and, for a given state $s_t$ in the sequence, the next state $s_{t+1}$ is sampled from $p(\cdot|s_t, a_t)$, where $a_t = \arg\max_a \pi(s_t, a)$. The use of the $\arg\max$ operator over $\pi$ reduces the noise in the selection process of the DEC-OPTIONS because the options also act greedily according to the sub-policies extracted from $\pi$. If $\rho$ is episodic, $s_{T+1}$ is a terminal state; $T + 1$ defines a maximum horizon for the sequence otherwise. $\mathcal{T}_i$ is a set of such sequences for $\pi_{\theta_i}$ and $\rho_i$'s $\mu$. Finally, $\mathcal{T} = \{\mathcal{T}_1, \mathcal{T}_2, \cdots, \mathcal{T}_n\}$.

The sub-policies $U$ do not offer temporal abstractions as they are executed only once. We turn these sub-policies into options by wrapping each $\pi$ in $U$ with a while-loop of $z$ iterations. Once the resulting option $\omega$ is initiated, it will run for $z$ steps before terminating. We denote as $\omega_z$ the $z$-value of $\omega$. In each iteration of the while loop, the agent will execute in the environment the action $\arg\max_a \pi(s, a)$, where $\pi$ is the sub-policy and $s$ is the agent's current state. The $\arg\max$ operator ensures that the policy is deterministic within the loop. Let $T_{\max}$ be the length of the longest sequence in $\mathcal{T}$, then we consider options with $z = 1, \cdots, T_{\max}$ for each sub-policy in $U$. Let $\Omega = \{\Omega_1, \Omega_2, \cdots, \Omega_n\}$ be the set of all while-loop options obtained from $U$. Each $\Omega_i$ has $T_{\max} \cdot |U_i|$ options for $\rho_i$, one for each $z$. Our task is to select a subset of "helpful" options from $\Omega$.

We measure whether a set of options is helpful in terms of the Levin loss (Orseau et al., 2018) of the set. The Levin loss measures the expected number of environmental steps (calls to the function $p$) an agent needs to perform with a given policy to reach a target state. The Levin loss assumes that $p$ is deterministic and the initial state is fixed; the only source of variability comes from the policy. The Levin loss for a sequence $\mathcal{T}_i$ and policy $\pi$ is $\mathcal{L}(\mathcal{T}_i, \pi) = \frac{|\mathcal{T}_i|}{\prod_{(s,a) \in \mathcal{T}_i} \pi(s, a)}$. The factor $1 / \prod_{(s,a) \in \mathcal{T}_i} \pi(s, a)$ is the expected number of sequences that the agent must sample with $\pi$ to observe $\mathcal{T}_i$. We assume that the length of the sequence the agent must sample to observe $\mathcal{T}_i$ is known to be $|\mathcal{T}_i|$ and therefore is fixed, so the agent performs exactly $|\mathcal{T}_i|$ steps in every sequence sampled.

Let $\pi_u$ be the uniform policy for an MDP, that is, a policy that assigns equal probability to all actions available in a given state. Furthermore, let $\pi_u^\Omega$ be the uniform policy when we augment the MDP actions with a set of options $\Omega$. There are two effects once we add a set of options to the set of available actions. First, the Levin loss can increase because the probability of choosing each action decreases, including the actions in the target sequence. Second, the Levin loss can decrease because the number of decisions the agent needs to make can also decrease, potentially reducing the number of multiplications performed in the denominator of the loss. Our task is then to select a subset of options from the set $\Omega$ generated with decomposed policies such that we minimize the Levin loss.

$$\arg\min_{\Omega' \subseteq \Omega_T} \sum_{\mathcal{T}_i \in \mathcal{T}_V} \mathcal{L}(\mathcal{T}_i, \pi_u^{\Omega'}). \tag{1}$$

We divide the set of tasks $\mathcal{P}$ into disjoint training and validation sets to increase the chances of selecting options that generalize. For example, an option that encodes $\pi_{\theta_i}$ with a loop that iterates for $z$ steps, where $z$ is equal to the length of the sequences in $\mathcal{T}_i$, is unlikely to generalize to other tasks, as it is highly specific to $\rho_i$. In Equation 1, $\Omega_T$ is the set of options extracted from the policies learned for the tasks in the training set and $\mathcal{T}_V$ are the sequences obtained by rolling out the policies learned for the tasks in the validation set. We consider uniform policies in our formulation because they

approximate neural policies in the first steps of training, since the network's weights are randomly initialized. By minimizing the Levin loss, we reduce the expected number of sequences that the agent samples to observe high reward values. Solving the subset selection problem in Equation 1 is NP-hard (Garey & Johnson, 1979), so we use a greedy approximation to solve the problem.

### 4.2.1 GREEDY APPROXIMATION TO SELECT OPTIONS

The greedy algorithm for approximating a solutions to Equation 1 initializes $\Omega'$ as an empty set and, in every iteration, adds to $\Omega'$ the option that represents the largest decrease in Levin loss. The process stops when adding another option does not decrease the loss, so it returns the subset $\Omega'$.

Due to the call-and-return model, we need to use a dynamic programming procedure to efficiently compute the values of $\mathcal{L}$ while selecting $\Omega'$. This is because it is not clear which action/option the agent would use in each state of a sequence so that the probability $\prod_{(s,a)\in\mathcal{T}_i} \pi(s,a)$ is maximized. For example, an option $\omega$ returns the correct action for $\omega_z$ states in the sequence starting in $s_1$. While $\omega'$ does not return $a_1$ in $s_1$, it returns the correct action in the sequence for $\omega'_z$ states from $s_2$. If $\omega_z < \omega'_z$ and using $\omega$ in $s_1$ prevented us from using $\omega'$ in $s_2$ because $\omega$ would still be executing in $s_2$, then using $a_1$ in $A$ for $s_1$ and then starting $\omega'$ in $s_2$ could maximize $\prod_{(s,a)\in\mathcal{T}_i} \pi(s,a)$.

---

**Algorithm 1** COMPUTE-LOSS

**Require:** Sequence $\mathcal{S} = \{s_0, s_1, \cdots, s_{T+1}\}$ of states of a trajectory $\mathcal{T}$, probability $p_{u,\Omega}$, options $\Omega$
**Ensure:** $\mathcal{L}(\mathcal{T}, \pi_u^\Omega)$
 1: $M[j] \leftarrow j$ for $j = 0, 1, \cdots, T+1$ *# initialize table assuming only primitive actions*
 2: **for** $j = 0$ to $T+1$ **do**
 3:    **for** $\omega$ in $\Omega$ **do**
 4:       **if** $\omega$ is applicable in $s_j$ **then**
 5:          $M[j + \omega_z] \leftarrow \min(M[j + \omega_z], M[j] + 1)$ *# $\omega$ is used in $s_j$ for $\omega_z$ steps*
 6:    **if** $j > 0$ **then**
 7:       $M[j] \leftarrow \min(M[j-1] + 1, M[j])$
 8: **return** $|\mathcal{T}| \cdot (p_{u,\Omega})^{-M[T+1]}$

---

Algorithm 1 shows the computation of $\mathcal{L}$. The procedure receives a sequence $\mathcal{S}$ of states from a trajectory $\mathcal{T}$; the sequence includes $\mathcal{T}$'s terminal state $s_{T+1}$. It also receives the probability $p_{u,\Omega} = \frac{1}{|A|+|\Omega|}$ of choosing any of the available actions under the uniform policy when the action space is augmented with the options in $\Omega$. Finally, it also receives the set of options $\Omega$ and returns $\mathcal{L}(\mathcal{T}, \pi_u^\Omega)$.

To compute the Levin loss, one needs to decide whether each transition $s_i$ to $s_{i+1}$ in $\mathcal{T}$ is covered by an action in $A$ or an option in $\Omega$. Algorithm 1 verifies all possibilities for each of these pairs such that the Levin loss is minimized. It uses a table $M$ with one entry for each state in $\mathcal{S}$, which stores the smallest number of decisions the agent must make to reach each state in the sequence. Initially, the procedure assumes that the agent reaches all states with primitive actions (line 1): $M[j] = j$ for all $j$. Then, it updates the entries of $M$ (lines 5 and 7) by verifying which options $\omega$ can be applied to each state. At the end of the computation (line 8), $M[j]$ stores the smallest number of decisions that the agent must make to reach $s_j$, for all $j$, including $T+1$. The minimum loss for $p_{u,\Omega}$ and $\Omega$ is based on the smallest number of decisions needed to reach the end of the sequence, $s_{T+1}$ (line 8).

Algorithm 1 computes the Levin loss for a set of options $\Omega$ in $O(|\Omega| \cdot T)$ time steps; its memory complexity is $O(T)$. We present in Appendix A an example of the computation of Algorithm 1.

## 5 EXPERIMENTS

We conducted experiments to evaluate the hypothesis that neural networks encoding policies for MDPs may contain useful underlying options that can be extracted through neural decomposition.

In our experiments, we consider a set of tasks $\mathcal{P}$, all of which share a common state representation and output structure, but may differ in terms of reward functions and dynamics. The primary objective is to evaluate an agent that uses an action space augmented with DEC-OPTIONS synthesized from $\mathcal{P}$ on a new set of tasks $\mathcal{P}'$. We use Proximal Policy Optimization (PPO) (Schulman et al.,

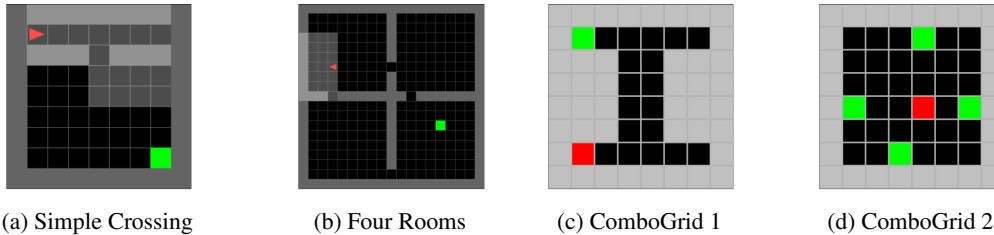

(a) Simple Crossing     (b) Four Rooms     (c) ComboGrid 1     (d) ComboGrid 2

Figure 2: Representative images of the problem domains used in our experiments.

2017) in the set of tasks $\mathcal{P}$, implemented using the Stable-baselines framework (Raffin et al., 2021). The policy network was configured as a small feed-forward neural network with ReLU neurons. We use small neural networks to be able to evaluate all sub-policies in the greedy selection step of DEC-OPTIONS. This way, we are able to evaluate our hypothesis without the added complexity of searching in the space of sub-policies. We use larger networks for the value function, as the value network is not used in DEC-OPTIONS. The task of learning policies for the set $\mathcal{P}'$ does not have to be solved with actor-critic algorithms. This is because the DEC-OPTIONS are used to augment the action space of the agents and any learning algorithm can be used. Thus, we evaluate both the Deep Q-Network (DQN) (Mnih et al., 2013) algorithm alongside PPO. Additional details, including agent architectures, hyperparameter settings, and used libraries are provided in the Appendix.

**Baselines** If the DEC-OPTIONS are helpful, then the option-augmented agent should be more sample-efficient than DQN and PPO operating in the original action space (**Vanilla-RL**). To demonstrate the importance of the decomposition and selection steps of DEC-OPTIONS, we also consider a baseline where the action space of the agents is augmented with the neural policies learned from the tasks $\mathcal{P}$ (**Neural-Augmented**). We also consider a variant of DEC-OPTIONS where we only perform the generation and selection steps, without performing decomposition, i.e., we consider the non-decomposed policies for $\mathcal{P}$ while generating options (**Dec-Options-Whole**). This comparison serves as evidence that neural decomposition facilitates transfer learning compared to using the complete policy. We also consider a baseline where both the value and policy models are trained in a task and initialized with the weights of the previously learned model when learning to solve a new task (**Transfer-PPO**). Since masking methods also access sub-policies of a network, we consider **Modulating-Mask** as a baseline. We use **PNN**, Progressive Neural Networks, as a representative baseline of methods that adapt the neural architecture. To demonstrate the impact of knowledge transfer, we consider **Option-Critic**, **ez-greedy**, and **DCEO** as representative algorithms for learning options directly in tasks $\mathcal{P}'$, without using knowledge of the policies learned for tasks in $\mathcal{P}$.

### 5.1 PROBLEM DOMAINS

We use two domains where exploration can be difficult and it is easy to generate similar but different tasks: MiniGrid (Chevalier-Boisvert et al., 2023) and ComboGrid, which we introduce in this paper.

**MiniGrid** The first domain is based on Minigrid environments. In MiniGrid, the agent is restricted to a partial observation of its surroundings, determined by a view size parameter. We select a set of simple tasks as set $\mathcal{P}$ and a set of more challenging tasks as set $\mathcal{P}'$. In $\mathcal{P}$, we use three instances of the Simple Crossing Environment with $9 \times 9$ grids and one wall, as shown in Figure 2a. For the test set $\mathcal{P}'$, we use three configurations of the Four Rooms Environment, as illustrated in Figure 2b. In Four Rooms, the agent navigates a $19 \times 19$ grid divided into four rooms. In the first task, the agent and the goal point are in the same room. In the second task, they are located in neighboring rooms, and in the third task, which is shown in Figure 2b, they are located in two non-neighboring rooms.

**ComboGrid** In this environment, the agent has full observational capabilities. The agent's movements are determined by a combination of actions (combo). Four unique combinations, each corresponding to a move to a neighboring cell, dictate the agent's navigation. If the agent successfully executes the correct series of actions corresponding to a valid combo, it moves to the corresponding neighboring cell. Failing to execute the correct combo results in the agent remaining in its current position and the history of previous combo actions being reset. The state contains not only the grid, but also the agent's current sequence of actions. A reward of $-1$ is assigned in the environment,

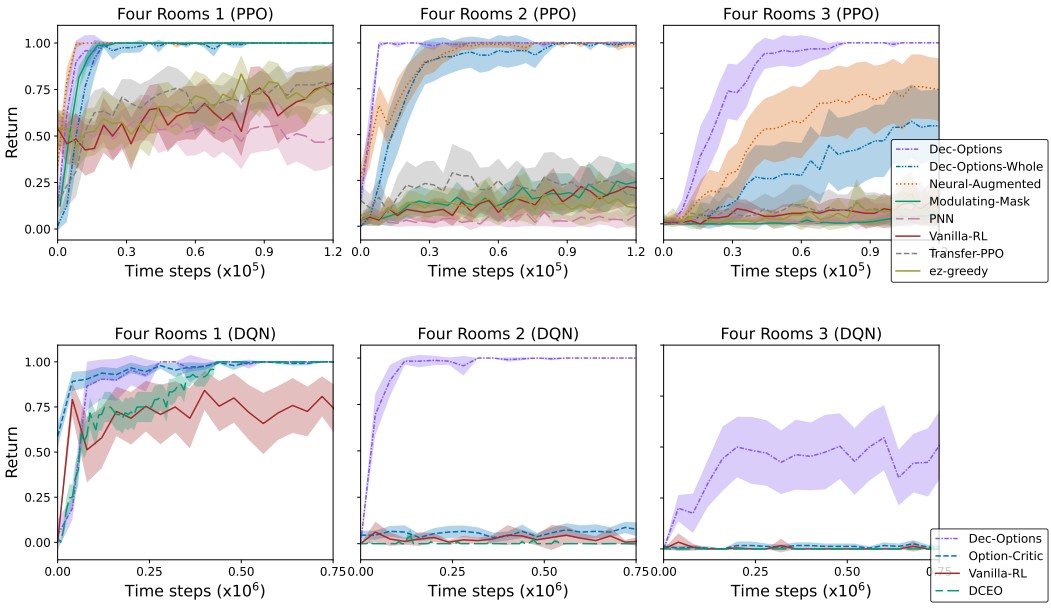

Figure 3: Performance of different methods on MiniGrid Domain.

with a reward of $0$ granted once the terminal point is reached. We evaluate DEC-OPTIONS and the baselines on four versions of ComboGrid, each differing in grid size: $(3 \times 3), (4 \times 4), (5 \times 5), (6 \times 6)$. For the set $\mathcal{P}$, we use four instances of each grid size; Figure 2c shows one of the tasks in $\mathcal{P}$ for size $6 \times 6$. Tasks in $\mathcal{P}$ differ in their initial and goal positions, which can cause interference for algorithms that do not decompose policies while transferring knowledge. Figure 2d shows the instance in $\mathcal{P}'$ for grid size $6 \times 6$. The agent receives a reward of $10$ for collecting each green marker.

## 5.2 EMPIRICAL RESULTS

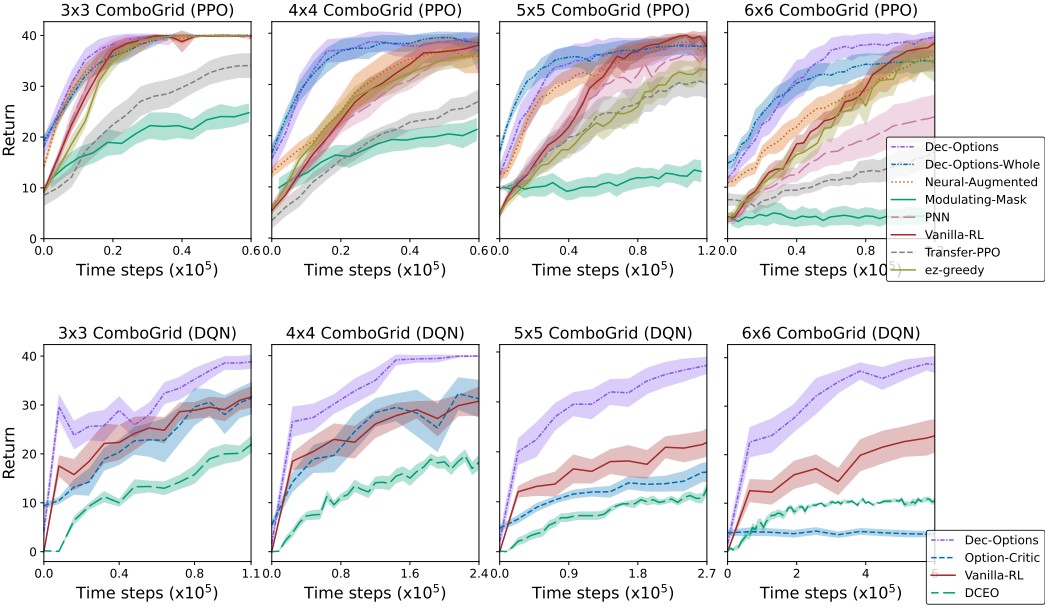

Figure 4: Performance of different methods on ComboGrid Domain.

Figure 3 shows the learning curves of agents equipped with DEC-OPTIONS and of the baselines. The first row shows the results for agents using PPO, while the second row shows the results for agents using DQN. Each plot shows the average return the agent receives over 24 independent runs of each system; the plots also show the 95% confidence interval of the runs. Since the domain is not discounted, the maximum reward is 1.0. Each instance of Four Room: 1, 2, and 3, represents a domain in which the agents using transfer learning attempt to learn after learning a policy for tasks in $\mathcal{P}$. Similarly to Figure 3, Figure 4 shows the results for the ComboGrid domain. Here, the maximum return the agent can obtain is 40, 10 for each marker the agent collects.

In Minigrid, Vanilla-RL fails to learn a good policy with the number of samples used in our experiments. The dynamics of the environment, where the agent needs to turn and then move, contributes to this difficulty. Transfer-PPO and PNN also faced difficulties due to interference caused by differences between tasks (Kessler et al., 2022). In contrast, DEC-OPTIONS, Dec-Options-Whole, Neural-Augmented, and Modulating-Mask performed well in Four Rooms 1. Regarding the DQN agents, Option-Critic, DCEO, and DEC-OPTIONS outperformed Vanilla-RL. As the complexity of the task increases, most of the baselines do not converge to an optimal policy. In particular, only the DEC-OPTIONS agent learned optimal policies in Four Rooms 3.

In the smallest ComboGrid, most of the baselines yield similar performance. Similarly to Minigrid, as we increase the size of the grid, we start to notice differences among the methods. For grids of size $4 \times 4$, DEC-OPTIONS and Dec-Options-Whole converge quicker to an optimal policy than the other methods. For grid of size $6 \times 6$, DEC-OPTIONS converges quicker than all other methods. For the DQN experiments, we also observe similar results to those observed in MiniGrid, where the gap between DEC-OPTIONS and other methods increases as the grid size increases.

Although we compare DEC-OPTIONS with other option-learning algorithms, we note that DEC-OPTIONS solves a fundamentally different problem than the other methods evaluated. For example, DCEO focuses on exploration while learning options for a target task. In contrast, we learn options on a set of tasks and evaluate whether these options are helpful in downstream problems. Future research might investigate the use of some of these baselines in the context of transfer learning.

**Examples** DEC-OPTIONS learned long options for MiniGrid. The sequence below shows the actions from state to goal (left to right) of the DEC-OPTIONS agent in the Four Room 3 environment. Here, 0, 1, and 2 mean 'turn right', 'turn left', and 'move forward', respectively. The curly brackets show what is covered by one of the options learned. The options reduce the number of agent decisions from 39 to only 10 in this example.

$$0, \underbrace{2,2,2,2,2,2,2,2,2,2,0,2,2}_{\text{Option 1}}, 1, \underbrace{0,2,2,2,2,2,2,2,2,2,2,2,0}_{\text{Option 1}}, 0, 2,2,2,2,2,0, \underbrace{0,2,2,2}_{\text{Option 1}}$$

We show a sub-sequence of an episode of the DEC-OPTIONS agent in the $4 \times 4$ ComboGrid. Option 2 learns sequences of actions that move the agent to another cell on the grid (e.g., "Down" and "Right"). Option 1 is shorter than Option 2 and it applies in many situations. For example, calling Option 1 twice can move the agent up, or even finish a left move to then move up.

$$\overset{\text{Down}}{\underbrace{0,2,2,1}_{\text{Option 2}}}, \overset{\text{Right}}{\underbrace{1,2,1,0}_{\text{Option 2}}}, \overset{\text{Up}}{\underbrace{0,0}_{\text{Option 1}}}, \overset{}{\underbrace{1,1}_{\text{Option 1}}}, \overset{\text{Left}}{\underbrace{1,0,2,2}_{\text{Option 2}}}, \overset{\text{Down}}{\underbrace{0,2,2,1}_{\text{Option 2}}}, \overset{\text{Down}}{\underbrace{0,2,2,1}_{\text{Option 2}}}, \overset{\text{Down}}{\underbrace{0,2,2,1}_{\text{Option 2}}}, 1, \overset{\text{Left}}{\underbrace{0,2}_{\text{Option 1}}}, \overset{}{\underbrace{2,0}_{\text{Option 1}}}, 0, \overset{\text{Up}}{\underbrace{1,1}_{\text{Option 1}}}$$

## 6 CONCLUSIONS

In this paper, we showed that options can "occur naturally" in neural network encoding policies and that we can extract such options through neural decomposition. We called the resulting options DEC-OPTIONS. Our decomposition approach assumes a set of related tasks where the goal is to transfer knowledge from one set of tasks to another. Since the number of options we can extract from neural networks grows exponentially with the number of neurons in the network, we use a greedy procedure to select a subset of them that minimizes the Levin loss. By minimizing the Levin loss, we increase the chances that the agent will apply sequences of actions that led to high-reward states in previous tasks. We evaluated our decomposition and selection approach on hard-exploration grid-world problems. The results showed that DEC-OPTIONS accelerated the learning process in a set of tasks that are similar to those used to train the models from which the options were extracted.

ACKNOWLEDGEMENTS

This research was supported by Canada's NSERC and the CIFAR AI Chairs program, and was enabled in part by support provided by the Digital Research Alliance of Canada. We thank the reviewers for the discussions and their suggestions, Marlos Machado for helpful comments on an early draft of this work, and Martin Klissarov for answering questions we had regarding DCEO.

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

## A    EXAMPLE OF THE ALGORITHM FOR COMPUTING THE LEVIN LOSS

Let $\mathcal{S} = \{s_0, s_1, s_2, s_3, s_4, s_5\}$ be a sequence of states and $\Omega = \{\omega_1, \omega_2\}$ be a set of options. $\omega_1$ can start in $s_0$ and it terminates in $s_2$; $\omega_2$ can start in $s_1$ and it terminates in $s_4$. Next, we show how the table $M$ in Algorithm 1 changes after every iteration of the for-loop in line 2.

| Iterations | $M$ |
|:----------:|:----:|
| initialization | 0, 1, 2, 3, 4, 5 |
| 0 | 0, 1, 1, 3, 4, 5 |
| 1 | 0, 1, 1, 3, 2, 5 |
| 2 | 0, 1, 1, 3, 2, 5 |
| 3 | 0, 1, 1, 2, 2, 5 |
| 4 | 0, 1, 1, 2, 2, 5 |
| 5 | 0, 1, 1, 2, 2, 3 |

The value of 3 of $M[5]$ at the end of the last iteration indicates that the state $s_5$ can be reached with three actions: a primitive action from $s_0$ to $s_1$, $\omega_2$ from $s_1$ to $s_4$, and another primitive action from $s_4$ to $s_5$. If $p_{u,\Omega} = 0.25$, then the optimal Levin loss value returned in line 8 of Algorithm 1 for $\mathcal{T}$ and $\Omega$ is $\frac{6}{0.25^3} = 384$.

## B    EXPERIMENTS DETAILS

### B.1    OPTIONS SELECTION PROCESS

In our methodology, we employ an approach to select options based on a set of sequences denoted $\mathcal{T} = \{\mathcal{T}_1, \mathcal{T}_2, \cdots, \mathcal{T}_n\}$, as described in Section 4.2. This selection process is driven by a greedy algorithm, facilitating the identification of options that can be "helpful". However, it is the case that when decomposing the policy $\pi_{\theta_i}$, the resulting sub-policies will have a low Levin loss on sequence $\mathcal{T}_i$. To mitigate this bias, we compute the Levin loss for sub-policies derived from the neural decomposition of $\pi_{\theta_i}$ on the set $\mathcal{T} \setminus \{\mathcal{T}_i\}$.

### B.2    PLOTS

Figures shown in Section 5 were generated using a standardized methodology. To ensure robustness, we applied a systematic procedure to all our baselines. We began the process by performing a hyperparameter search, as outlined in B.3, to select the best hyperparameters for each baseline. Subsequently, we perform 30 independent runs (seeds) for each baseline. We discarded the 20% of the independent runs with the poorest performance. We computed the mean and 95% confidence intervals over the remaining 24 seeds.

### B.3    ARCHITECTURE AND PARAMETER SEARCH

For algorithms utilizing the Proximal Policy Optimization (PPO) framework, including **Vanilla-RL PPO**, **Neural-Augmented**, **Transfer-PPO**, **PNN**, **ez-greedy**, **Dec-Options-Whole**, and **Dec-Options** baselines, we used the stable-baselines library (Raffin et al., 2021). A comprehensive parameter search was conducted, encompassing the clipping parameter, entropy coefficient, and learning rate. These parameters are reported for each domain in their respective sections. In the case of **PNN**, we leveraged the library provided in `https://github.com/arcosin/Doric`, while still relying on the PPO algorithm from stable-baselines for training. For the **ez-greedy** algorithm, we integrated the temporally-extended $\varepsilon$-greedy algorithm with the PPO algorithm from the stable-baselines. We set the $\varepsilon$ to be equal to 0.01 and the $\mu$ to be equal to 2, as ez-greedy's original work. For **Option-Critic** implementations, we used the implementation in `https://github.com/lweitkamp/option-critic-pytorch` and the best parameters found for the learning rate and the number of options in a hyperparameter sweep process. For the **DCEO** baseline, we used the implementation from the original paper (`https://github.com/mklissa/dceo/`).

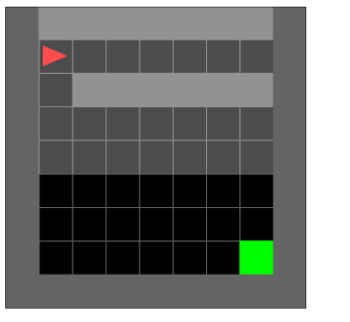 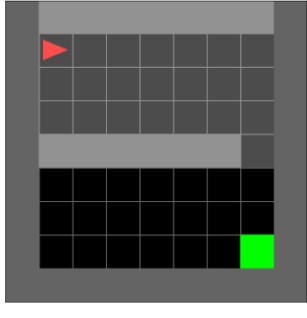 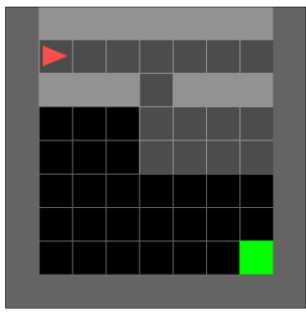

(a) Simple Crossing - Seed 0      (b) Simple Crossing - Seed 1      (c) Simple Crossing - Seed 2

Figure 5: MiniGrid Simple Cross Tasks

The parameter search is then applied to the learning rate, the number of options, and the probability of executing an option. As for the **Modulating-Mask** method, we used the code provided by the original authors (`https://github.com/dlpbc/mask-lrl`). In scenarios where the Deep Q-Network (DQN) was employed, such as **Vanilla-RL DQN** and **Dec-Options DQN** baselines, we adhered to the stable-baselines. Similarly to PPO, we performed a parameter search, this time targeting Tau and the learning rate, while keeping other parameters fixed. All the parameter searches mentioned were performed using the grid search method (LaValle et al., 2004).

For tasks within the MiniGrid domain, we employed a feedforward network. The policy network structure consisted of a single hidden layer comprising 6 nodes, while the value network used three hidden layers with 256 neurons each. As tasks transitioned to $\mathcal{P}'$, we expanded the policy network to encompass three hidden layers with 50 neurons in each layer. In the **Transfer-PPO** method, we also used a policy network with three hidden layers with 50 neurons in each across all tasks in both $\mathcal{P}$ and $\mathcal{P}'$. Regarding the **Modulating-Mask** approach, we retained the structure of the neural network as in its original implementation. This structure featured a shared feature network with three hidden layers, each with 200 neurons, followed by one hidden layer with 200 neurons for the policy network and another hidden layer also with 200 neurons for the value network. For DQN-based methods, including **Vanilla-RL DQN** and **Dec-Options DQN**, as well as **Option-Critic** and **DCEO**, the neural network had three hidden layers, each with 200 neurons.

In the ComboGrid domain, the architectural and parameter search aspects mirrored those of the MiniGrid domain for DEC-OPTIONS. The policy network structure featured one hidden layer with 6 nodes for tasks in $\mathcal{P}$, and we increased it to 16 nodes for tasks in $\mathcal{P}'$. **Transfer-PPO** maintained a uniform policy network structure of one hidden layer with 16 nodes across all tasks in $\mathcal{P}$ and $\mathcal{P}'$. The value network is a network with 3 hidden layers with 200 neurons each for tasks in $\mathcal{P}$ and $\mathcal{P}'$. As for the **Modulating-Mask** method, the structure of the neural network is as before. For DQN-based methods and **Option-Critic**, we adopted a two-layer configuration with 32 neurons in the first layer and 64 neurons in the second hidden layer.

## C  DOMAINS

### C.1  MINIGRID

As described in Section 5, we leveraged the MiniGrid implementation provided by Chevalier-Boisvert et al. (2023). For our experiments, we opted for the Simple Crossing tasks, which formed the training set denoted as $\mathcal{P}$. Notably, we selected three different variants of 'MiniGrid-SimpleCrossingS9N1-v0', as illustrated in Figure 5. For the test set $\mathcal{P}'$, we chose three versions of the Four Rooms domain, as depicted in Figure 6. In these environments, actions for interacting with object are not needed and were not available to the agent. The agent had access to three primitive actions: 'Action Left' (represented by 0) for left turns, 'Action Right' (represented by 1) for right turns, and 'Action Forward' (represented by 2) for forward movement. At each step, the agent views a small field in front of itself, making the environment partially observable. Also, its view is obstructed by walls where applicable. Regarding the representation of the state, we employed

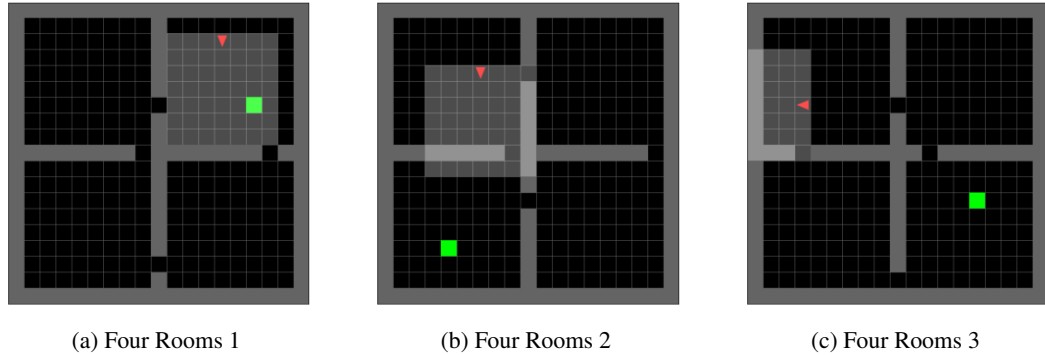

(a) Four Rooms 1         (b) Four Rooms 2         (c) Four Rooms 3

Figure 6: MiniGrid Four Rooms Tasks

a modified version of the original MiniGrid observation. At each time step, the agent's view is constrained to a 5×5 grid in front of itself. This observation was then transformed into a one-hot encoding format. The one-hot encoding encapsulated various elements, including the goal, walls, and empty floor spaces. Additionally, we incorporated the agent's directional orientation into the observation, consistent with the conventions of the original MiniGrid. This directional indicator served as a compass for the agent, aiding it in determining its current facing direction. This representation aimed to streamline observations without altering the fundamental dynamics of the original library. To maintain consistency across our experiments, we set a maximum episode length of 1000 for tasks in set $\mathcal{P}$ and 361 for tasks in set $\mathcal{P}'$. In terms of rewards, we assigned a value of -1 for each step in tasks within set $\mathcal{P}$, except at termination points. Tasks in set $\mathcal{P}'$ were designed with rewards of 0 for all states, except upon reaching the terminal state, where a reward of 1 was granted.

**Parameter Search** In pursuit of reasonable hyperparameters, we performed a parameter search for all methods used in our experiments. We have outlined the search spaces for each method below.

For methods using the PPO algorithm, we explored the following hyperparameter ranges:

- **Learning Rate:** 0.005, 0.001, 0.0005, 0.0001, 0.00005
- **Clipping Parameter:** 0.1, 0.15, 0.2, 0.25, 0.3
- **Entropy Coefficient:** 0.0, 0.05, 0.1, 0.15, 0.2

For methods relying on the Deep Q-Network (DQN) paradigm, our parameter search encompassed the following hyperparameter lists:

- **Learning Rate:** 0.01, 0.005, 0.001, 0.0005, 0.0001
- **Tau:** 1., 0.7, 0.4, 0.1

In the case of the Option-Critic method, we explored the following hyperparameter combinations:

- **Learning Rate:** 0.01, 0.005, 0.001, 0.0005, 0.0001, 0.00005
- **Number of Options** 2, 3, 4, 5, 6

For the DCEO baseline, we explored the following hyperparameter combinations:

- **Learning Rate:** 0.01, 0.005, 0.001, 0.0005, 0.0001, 0.00005
- **Number of Options:** 3, 5, 10
- **Option Probability:** 0.2, 0.7, 0.9

The hyperparameter values used in our experiments with MiniGrid problems are given in Tables 1, 2, 3, 4, 5, and 6.

|  | Clipping Parameter | Entropy Coefficient | Learning Rate |
|---|---|---|---|
| **Vanilla-RL** | 0.15 | 0.05 | 0.0005 |
| **Neural-Augmented** | 0.2 | 0.2 | 0.0005 |
| **Transfer-PPO** | 0.15 | 0.15 | 0.0001 |
| **PNN** | 0.1 | 0.0 | 0.0001 |
| **Modulating-Mask** | 0.15 | 0.1 | 0.001 |
| **ez-greedy** | 0.15 | 0.05 | 0.0005 |
| **Dec-Options-Whole** | 0.3 | 0.15 | 0.0005 |
| **Dec-Options** | 0.25 | 0.1 | 0.0005 |

Table 1: Four Rooms 1 - PPO

|  | Tau/Number of Options | Learning Rate |
|---|---|---|
| **Vanilla-RL** | 0.7 | 0.0001 |
| **Option-critic** | 2 | 0.0001 |
| **Dec-Options** | 0.7 | 0.0005 |
| **DCEO** | 5 (Probability: 0.9) | 0.0005 |

Table 2: Four Rooms 1 - DQN

|  | Clipping Parameter | Entropy Coefficient | Learning Rate |
|---|---|---|---|
| **Vanilla-RL** | 0.1 | 0.2 | 0.0005 |
| **Neural-Augmented** | 0.15 | 0.0 | 0.0005 |
| **Transfer-PPO** | 0.1 | 0.05 | 5e-05 |
| **PNN** | 0.25 | 0.0 | 0.005 |
| **Modulating-Mask** | 0.2 | 0.0 | 0.01 |
| **ez-greedy** | 0.1 | 0.0 | 0.0001 |
| **Dec-Options-Whole** | 0.25 | 0.05 | 0.0005 |
| **Dec-Options** | 0.2 | 0.1 | 0.001 |

Table 3: Four Rooms 2 - PPO

|  | Tau/Number of Options | Learning Rate |
|---|---|---|
| **Vanilla-RL** | 1.0 | 0.0005 |
| **Option-critic** | 2 | 0.0001 |
| **Dec-Options** | 1.0 | 0.001 |
| **DCEO** | 10 (Probability: 0.9) | 0.001 |

Table 4: Four Rooms 2 - DQN

## C.2 COMBOGRID

In the ComboGrid environment, the agent's movement is dictated by a sequence of actions, which we refer to as "combos". These action sequences are described in the list below, for Down, Up, Right, and Left.

- Moving **Down**: 0, 2, 2, 1
- Moving **Up**: 0, 0, 1, 1
- Moving **Right**: 1, 2, 1, 0
- Moving **Left**: 1, 0, 2, 2

|  | Clipping Parameter | Entropy Coefficient | Learning Rate |
|---|---|---|---|
| **Vanilla-RL** | 0.2 | 0.0 | 5e-05 |
| **Neural-Augmented** | 0.1 | 0.0 | 0.001 |
| **Transfer-PPO** | 0.1 | 0.05 | 0.0001 |
| **PNN** | 0.15 | 0.0 | 0.001 |
| **Modulating-Mask** | 0.1 | 0.0 | 0.005 |
| **ez-greedy** | 0.25 | 0.05 | 0.0005 |
| **Dec-Options-Whole** | 0.15 | 0.05 | 0.001 |
| **Dec-Options** | 0.2 | 0.1 | 0.001 |

Table 5: Four Rooms 3 - PPO

|  | Tau/Number of Options | Learning Rate |
|---|---|---|
| **Vanilla-RL** | 0.7 | 0.001 |
| **Option-critic** | 3 | 0.0001 |
| **Dec-Options** | 0.1 | 0.0005 |
| **DCEO** | 10 (Probability: 0.9) | 0.0001 |

Table 6: Four Rooms 3 - DQN

In our experiments, we conducted tests under four settings of the ComboGrid environment, each varying in size. The tasks in $\mathcal{P}$, of sizes $5 \times 5$ and $6 \times 6$, are shown in Figure 7. Across all tasks in the set $\mathcal{P}$, we established a uniform reward structure, with a value of -1 assigned for every step taken in the environment, except when reaching the terminal state, where the reward was set to 0. The tasks in set $\mathcal{P}'$ for different grid sizes are also presented in Figure 8. These tasks adhered to a different reward structure, with a reward value of 0 assigned for all states except those corresponding to goal points, where the reward was set to 10. There were four distinct goal points on the grid, which added to the maximum cumulative reward attainable in a single episode of 40. The action space and the fully observable state space remained fixed for all tasks, regardless of the size of the grid, encompassing tasks in both sets, $\mathcal{P}$ and $\mathcal{P}'$.

Agents are presented with primitive actions represented by integers 0, 1, and 2, resulting in the creation of combinations ("combos") according to the specifications described above. For example, to move down, the agent needs to perform the sequence 0, 2, 2, 1. At each time step, the agent is provided with a complete view of the grid in the form of a one-hot representation. This representation encapsulated the position of the agent as well as the positions of the goals within the grid. Consequently, the dimension of the observation space was proportional to the size of the grid, resulting in a representation of size $W^2 \times 2$, where $W$ is the size of the grid. Additionally, the agent received the sequence of past combo actions, also in a one-hot encoding format, as part of its input. We enforce a maximum episode length of $W^2 \times 80$ steps for tasks in set $\mathcal{P}$ and $W^2 \times 16$ steps for tasks within set $\mathcal{P}'$.

**Parameter Search**  Similarly to MiniGrid, we performed a parameter search for all methods used in our experiments. We have outlined the search spaces for each method below.

For methods utilizing the PPO algorithm, we explored the following hyperparameter values.

- **Learning Rate:** 0.05, 0.01, 0.005, 0.001
- **Clipping Parameter** 0.05, 0.1, 0.15, 0.2, 0.25, 0.3
- **Entropy Coefficient:** 0.0, 0.05, 0.1, 0.15, 0.2

For methods relying on the Deep Q-Network (DQN) paradigm, our parameter search encompassed the following hyperparameter values.

- **Learning Rate:** 0.05, 0.01, 0.005, 0.001, 0.0005, 0.0001
- **Tau:** 1., 0.7, 0.4, 0.1

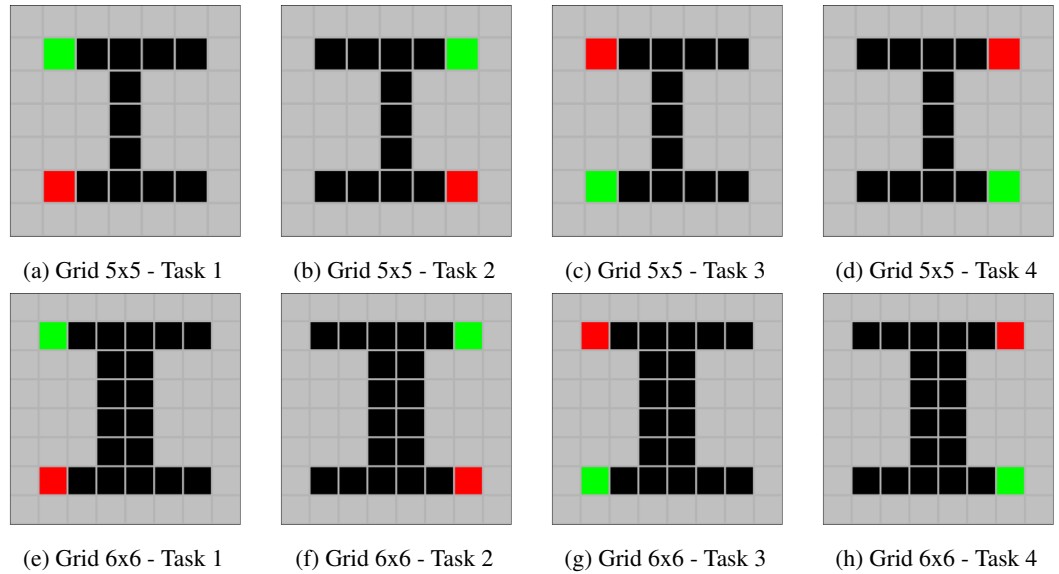

Figure 7: ComboGrid tasks. In this depiction, the agent is highlighted in red, while the goals are denoted in green, and the walls are represented in grey.

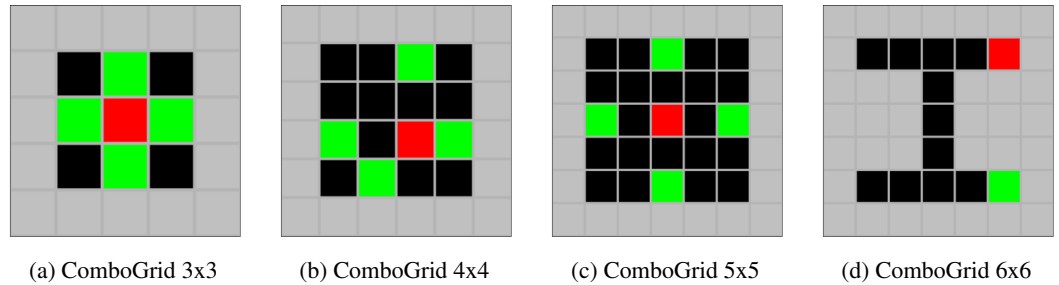

Figure 8: ComboGrid $\mathcal{P}'$ tasks

In the case of the Option-Critic method, we explored the following hyperparameter values.

- **Learning Rate:** 0.01, 0.005, 0.001, 0.0005, 0.0001, 0.00005
- **Number of Options** 2, 3, 4, 5, 6

For the DCEO baseline, we explored the following hyperparameter values.

- **Learning Rate:** 0.01, 0.005, 0.001, 0.0005, 0.0001
- **Number of Options:** 3, 5, 10
- **Option Probability:** 0.2, 0.7, 0.9

As for the Modulating-Mask method, we explored the following hyperparameters values.

- **Learning Rate:** 0.005, 0.001, 0.0005, 0.0001, 0.00005
- **Clipping Parameter** 0.1, 0.15, 0.2, 0.25
- **Entropy Coefficient:** 0.0, 0.05, 0.1, 0.15, 0.2

The hyperparameter values used in our experiments with ComboGrid problems are given in Tables 7, 8, 9, 10, 11, 12, 13, and 14.

|  | Clipping Parameter | Entropy Coefficient | Learning Rate |
|---|---|---|---|
| **Vanilla-RL** | 0.15 | 0.1 | 0.01 |
| **Neural-Augmented** | 0.25 | 0.05 | 0.01 |
| **Transfer-PPO** | 0.3 | 0.05 | 0.001 |
| **PNN** | 0.2 | 0.1 | 0.01 |
| **Modulating-Mask** | 0.1 | 0.1 | 0.0005 |
| **ez-greedy** | 0.15 | 0.05 | 0.005 |
| **Dec-Options-Whole** | 0.15 | 0.05 | 0.005 |
| **Dec-Options** | 0.2 | 0.05 | 0.005 |

Table 7: ComboGrid 3x3 - PPO

|  | Tau/Number of Options | Learning Rate |
|---|---|---|
| **Vanilla-RL** | 1. | 0.001 |
| **Option-critic** | 2 | 0.001 |
| **Dec-Options** | 1. | 0.001 |
| **DCEO** | 10 (Probability: 0.2) | 0.0005 |

Table 8: ComboGrid 3x3 - DQN

|  | Clipping Parameter | Entropy Coefficient | Learning Rate |
|---|---|---|---|
| **Vanilla-RL** | 0.1 | 0.0 | 0.005 |
| **Neural-Augmented** | 0.3 | 0.0 | 0.005 |
| **Transfer-PPO** | 0.05 | 0.2 | 0.001 |
| **PNN** | 0.2 | 0.1 | 0.005 |
| **Modulating-Mask** | 0.25 | 0.15 | 0.0001 |
| **ez-greedy** | 0.2 | 0.05 | 0.005 |
| **Dec-Options-Whole** | 0.25 | 0.05 | 0.01 |
| **Dec-Options** | 0.25 | 0.0 | 0.005 |

Table 9: ComboGrid 4x4 - PPO

|  | Tau/Number of Options | Learning Rate |
|---|---|---|
| **Vanilla-RL** | 0.7 | 0.0005 |
| **Option-critic** | 3 | 0.0005 |
| **Dec-Options** | 1. | 0.0005 |
| **DCEO** | 3 (Probability: 0.2) | 0.0005 |

Table 10: ComboGrid 4x4 - DQN

# D QUALITATIVE ANALYSIS OF RESULTS

## D.1 MINIGRID

In our study, we used heatmaps to visually represent the distribution of cells visited within the MiniGrid environment during training. These heatmaps provide information on the states that the agent frequents at different training steps. By comparing the heatmaps of the DEC-OPTIONS method with those of Vanilla-RL in Figure 9, we can see different exploration patterns, which can be related to the performance of the methods shown in Figure 3.

|  | Clipping Parameter | Entropy Coefficient | Learning Rate |
|---|---|---|---|
| **Vanilla-RL** | 0.25 | 0.1 | 0.005 |
| **Neural-Augmented** | 0.15 | 0.0 | 0.005 |
| **Transfer-PPO** | 0.1 | 0.2 | 0.001 |
| **PNN** | 0.25 | 0.05 | 0.005 |
| **Modulating-Mask** | 0.2 | 0.05 | 0.001 |
| **ez-greedy** | 0.2 | 0.15 | 0.005 |
| **Dec-Options-Whole** | 0.2 | 0.05 | 0.005 |
| **Dec-Options** | 0.2 | 0.05 | 0.005 |

Table 11: ComboGrid 5x5 - PPO

|  | Tau/Number of Options | Learning Rate |
|---|---|---|
| **Vanilla-RL** | 0.7 | 0.001 |
| **Option-critic** | 4 | 0.0001 |
| **Dec-Options** | 1. | 0.001 |
| **DCEO** | 5 (Probability: 0.7) | 0.001 |

Table 12: ComboGrid 5x5 - DQN

|  | Clipping Parameter | Entropy Coefficient | Learning Rate |
|---|---|---|---|
| **Vanilla-RL** | 0.1 | 0.05 | 0.005 |
| **Neural-Augmented** | 0.2 | 0.05 | 0.005 |
| **Transfer-PPO** | 0.3 | 0.05 | 0.001 |
| **PNN** | 0.05 | 0.0 | 0.005 |
| **Modulating-Mask** | 0.15 | 0.0 | 0.005 |
| **ez-greedy** | 0.2 | 0.05 | 0.005 |
| **Dec-Options-Whole** | 0.2 | 0.0 | 0.001 |
| **Dec-Options** | 0.15 | 0.05 | 0.005 |

Table 13: ComboGrid 6x6 - PPO

|  | Tau/Number of Options | Learning Rate |
|---|---|---|
| **Vanilla-RL** | 0.7 | 0.001 |
| **Option-critic** | 2 | 0.005 |
| **Dec-Options** | 0.7 | 0.001 |
| **DCEO** | 3 (Probability: 0.9) | 0.0001 |

Table 14: ComboGrid 6x6 - DQN

## D.2    COMBOGRID

To illustrate the DEC-OPTIONS learned in the ComboGrid problems, we sampled the trajectories of the policy trained to use the options. The following sequences are partial trajectories of the agents that highlight the use of the learned options. We present trajectories on ComboGrid problems of sizes $3 \times 3$, $5 \times 5$, and $6 \times 6$ (trajectories for the grids of size $4 \times 4$ are already presented in the main text). We present only partial trajectories for grids of sizes $5 \times 5$ and $6 \times 6$ to improve readability.

In grids of size $3 \times 3$, the learned options almost exactly match the dynamics of the problem. The exception is Option 4, which only partially executes the sequence of actions that allow the agent to move left. For grids of size $5 \times 5$ and $6 \times 6$, DEC-OPTIONS learns longer options. For example, for $5 \times 5$, Option 2 can almost complete the sequence of actions to go up twice, while for $6 \times 6$, Option 2 can perform the sequence of actions to go left twice.

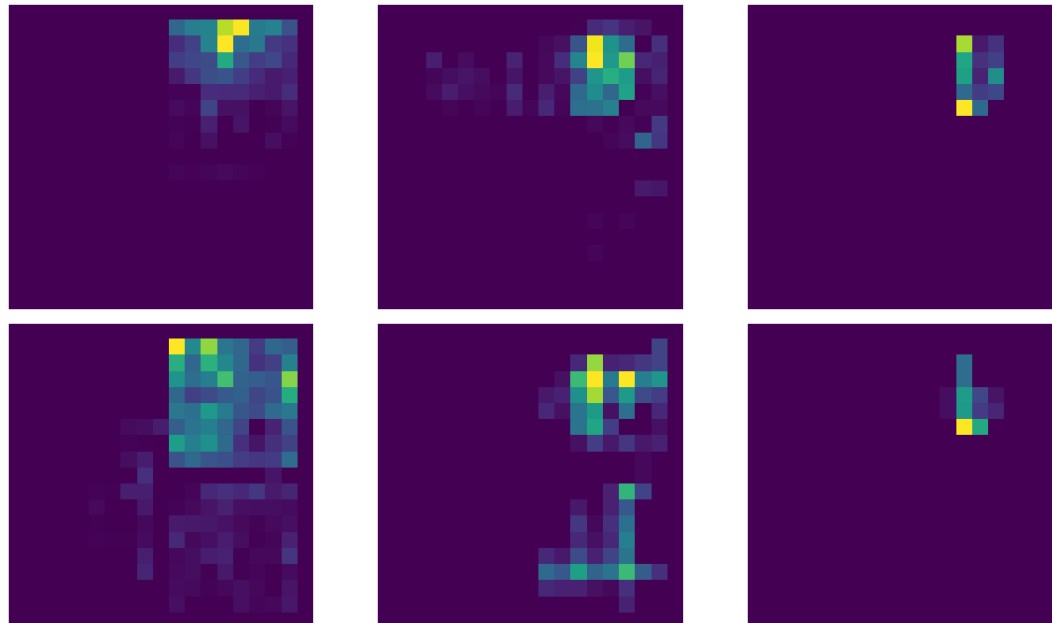

Figure 9: This figure shows the heatmap of cells visited in the Four Rooms 1 environment for both the DEC-OPTIONS and Vanilla-RL methods. The first row shows the heatmap of the DEC-OPTIONS's method, while the second row shows the Vanilla-RL method. The first, second, and third columns show the heatmap after 1, 3200, and 12800 steps, respectively.

**ComboGrid 3x3 (full trajectory):**

$$\underbrace{0,2,2,1}_{\text{Option 3}}\overbrace{}^{\text{Down}}, \underbrace{0,0,1,1}_{\text{Option 2}}\overbrace{}^{\text{Up}}, \underbrace{0,0,1,1}_{\text{Option 2}}\overbrace{}^{\text{Up}}, \underbrace{1,2,1,0}_{\text{Option 2}}\overbrace{}^{\text{Right}}, \underbrace{0,2,2,1}_{\text{Option 4}}\overbrace{}^{\text{Down}}, \underbrace{1,0,2,2}_{\text{Option 2}}\overbrace{}^{\text{Left}}, \underbrace{1,0,2,2}_{\text{Option 4}}\overbrace{}^{\text{Left}}$$

**ComboGrid 5x5:**

$$\underbrace{0,0,1,1,0,0,1,}_{\text{Option 2}}\overbrace{}^{\text{Up}\quad\text{Up}} \underbrace{1,1}_{\text{Option 1}}\overbrace{}^{}, \underbrace{2,1,0,}_{\text{Option 5}}\overbrace{}^{\text{Right}} \underbrace{1,2,1,}_{\text{Option 5}}\overbrace{}^{\text{Right}} \underbrace{0,0}_{\text{Option 4}}, \underbrace{2,2}_{\text{Option 4}}\overbrace{}^{\text{Down}}, \underbrace{1,0}_{\text{Option 4}}, \underbrace{2,2}_{\text{Option 3}}\overbrace{}^{\text{Down}}, 1$$

**ComboGrid 6x6:**

$$\underbrace{1,0,2,2,}_{\text{Option 2}}\overbrace{}^{\text{Left}\quad\text{Left}} \underbrace{1,0,2,2,}_{} \underbrace{0,0}_{\text{Option 3}}\overbrace{}^{\text{Up}}, \underbrace{1,1}_{\text{Option 1}}, \underbrace{0,0}_{\text{Option 3}}\overbrace{}^{\text{Up}}, \underbrace{1,1}_{\text{Option 1}}, \underbrace{0,0}_{\text{Option 3}}\overbrace{}^{\text{Up}}, \underbrace{1,1}_{\text{Option 1}}, \underbrace{1,2,1,0}_{\text{Option 4}}\overbrace{}^{\text{Right}}$$

## E  BASELINE SELECTING RANDOM OPTIONS

We also considered a baseline where we ignore steps 1 and 2 of DEC-OPTIONS and randomly select an option that performs a fixed set of actions of length $M$. We then add $K$ of such options to the agent's action set. Note that this approach requires domain knowledge, as the values of $M$ and $K$ are not known a priori. By contrast, DEC-OPTIONS discovers them automatically.

We ran experiments on the Combo domain where we chose $M$ to be 6, which is the average number of options DEC-OPTIONS chooses and $K$ to be 4, which is the number of directions the agent can move. We present the results in Figure 10. DEC-OPTIONS performs better than this baseline. Even when we manually set the values of $M$ and $K$, there are many options to choose from, and it is unlikely that helpful options would be selected this way.

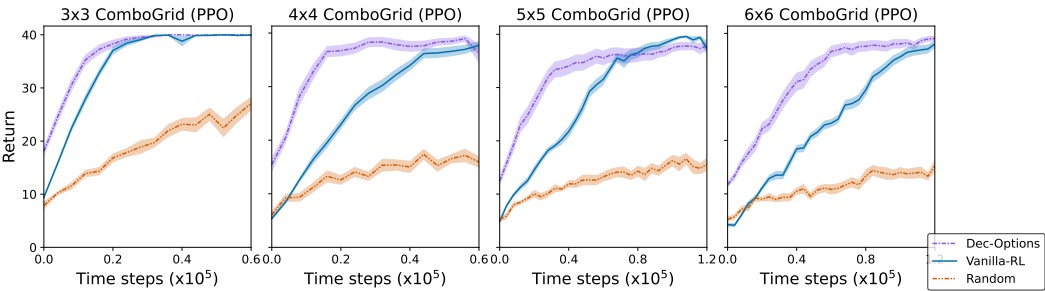

Figure 10: Comparison of Dec-Options with Vanilla-RL, and the Random baseline.

