# OpenReview forum: "Unveiling Options with Neural Network Decomposition"
_ICLR.cc/2024/Conference — ICLR 2024 poster_

### Official Review · Reviewer_voTZ · 2023-10-26

**Soundness:** 3 good
**Presentation:** 3 good
**Contribution:** 2 fair
**Rating:** 6
**Confidence:** 3

**Summary:**

The paper proposes to distill policies from a learned neural network and use these as options. They provide an algorithm that translates the neural network into a tree, decomposes it into sub-policies and selects their subset. In experiments, they verify the usefulness of the approach in several variants of grid-based domains.

**Strengths:**

The paper is well-written and is original in the way the options are created -- i.e., extracting them from an already trained neural network. However, the suggested approach is limited only to simple NNs.

**Weaknesses:**

First substantial weakness is that the number of sub-policies raises exponentially with the size of the NN, limiting this approach to very small networks and thus the significance of the proposed algorithm.

Second, given the large number of the subpolicies, it is not clear whether the subset selection (step 3) is better than selecting from a set of random policies. That is, if steps (1) and (2) are ommited and step (3) would select from a random set. In the second example at the end of Experiments, the subpolicies consist of 4 actions, which can be easily randomly generated. This would drastically reduce the complexity of the algorithm, but still would be limited to simle domains. This ablation is crucially missing from the evaluation.

**Questions:**

Please, present a definite argument (e.g., an experiment or a disproval why I am wrong) that the steps (1) and (2) are needed and the set of extracted sub-policies is useful, compared to a set of random sub-policies. I may reconsider the rating based on the arguments presented.

I assume that in the beginning of the section 4.2, the authors meant `where a_t = argmax_a \pi(s_t , a)`, instead of `where a_t = argmax_a p(s_t , a)`. However, if that is true, why not to sample the actions as `a_t ~ \pi(s)`? If this is to make a unique sequence per each policy, clearly indicate it for the reader and add a reason.

---
Suggestions:
- The graph in Figure 1-right confused me for a while, as I was trying to read it from left-to-right, instead of top-to-down. Indicate this fact for the reader. Also, consider (I don't stress this point) showing the complete graph with the path highlighted.

Typos:
- p. 7: Since learning a policy [IN] the new set ...
- p. 7: baseline Dec-Options-Whole[.]

---

> ### Author Response · Authors · 2023-11-21
>
> **Small Neural Networks**
>
> Dec-Options do not have to be restricted to small neural networks. Please see our answer to this question in the answer to all reviewers.
>
> **Baseline with Random Policies**
>
> Thank you for suggesting this baseline! In this baseline, we ignore steps 1 and 2 and randomly select an option that performs a fixed set of actions of length $M$. We then add $K$ of such options to the agent's action set. Note that this approach requires domain knowledge, as the values of $M$ and $K$ are not known a priori. By contrast, Dec-Options discovers them automatically.
>
> We ran experiments on the Combo domain where we chose $M$ to be $6$, which is the average number of options Dec-Option chooses and $K$ to be $4$, which is the number of directions the agent can move. Despite having more information than Dec-Options, the latter performs much better as shown in Figure 10 of the Appendix E of the revised version. Even when we manually set the values of $M$ and $K$, there are many options to choose from and it is unlikely that helpful options would be selected this way.
>
> Another interpretation of the reviewer's suggestion is to use the selection step (step 3) without steps 1 and 2. We note that step 3 cannot be used without step 1. This is because step 3 requires the data generated in step 1 to compute the Levin loss for selection.
>
> **Argmax in Section 4.2**
>
> Well spotted! Thank you! It should be $a_t = \arg\max_a \pi(s_t, a)$ instead of $a_t = \arg\max_a p(s_t, a)$. The use of the $\arg\max$ operator over $\pi$ reduces the noise in the selection process of the Dec-Options because the options also act greedily according to the sub-policies extracted from $\pi$. The added a sentence to explain this in the paper.
>
> **Suggestion Related to Figure 1**
>
> Thank you for your suggestion! We replaced Figure 1 with a smaller example where we can show the entire neural tree. This change also allowed us to get rid of the equations from the figure. Please see the new example on Page 4 of the revised version of the paper.

---

> > ### Comment · Reviewer_voTZ · 2023-11-21
> >
> > Dear authors, thank you for your answer.
> >
> > Regarding the baseline - please explain to me, why is not possible to compute the Levin loss with randomly generated policies. E.g., you randomly generate the policies, sample their trajectories and use Alg. 1.
> >
> > Thank you for adding additional baselines. However, Option-Critic, ez-greedy, and DCEO are not explained and a reader would be forced to read their papers. I acknowledge the lack of space, so maybe a brief explanation in appendix would be possible?
> >
> > Good job on Fig. 1, it is now much clearer.

---

> ### Author Response · Authors · 2023-11-21
>
> Thank you for your quick response. We are happy to hear you like our new Figure 1. We will add paragraphs explaining Option-Critic, DCEO, and $\epsilon$z-greedy to the appendix.
>
> In the meantime, we would like to ask a couple of clarifying questions about your suggested baseline. There is a chance we will also answer your question *"why is not possible to compute the Levin loss with randomly generated policies"* while asking our questions.
>
> **Our Question:** By randomly generated policies do you mean neural networks with their weights randomly generated or policies that execute a fixed and repeated random set of actions as we used in our baseline Random in Appendix E?
>
> **Re:** *"you randomly generate the policies, sample their trajectories and use Alg. 1"*
>
> There are a few ways of how the randomly generated policies $R$ can be used in Algorithm 1. Recall that Algorithm 1 requires a trajectory $\mathcal{T}$ and a set of options $\Omega$.
>
> 1. **Interpretation (a):** We can use $R$ to generate a trajectory $\mathcal{T}$ that is used in Algorithm 1.
> 2. **Interpretation (b):** We can use a set of policies $R$ to be used as the set of options $\Omega$ in Algorithm 1.
> 3. **Interpretation (c):** We can use policies $R$ to replace both $\mathcal{T}$ and $\Omega$ in Algorithm 1.
>
> **Interpretation (a)** still requires us to define the set of options $\Omega$ used in Algorithm 1. In Dec-Options, $\Omega$ is defined as the sub-policies of trained neural networks. If we use with the baseline the same set of options $\Omega$ used with Dec-Options, then we cannot skip steps 1 and 2 of Dec-Options because $\Omega$ is generated in those two steps.
>
> **Interpretation (b)** still requires us to define the trajectories $\mathcal{T}$ of Algorithms 1. In Dec-Options, $\mathcal{T}$ is given by trajectories sampled from fully trained neural networks, which are given in step 1 of our algorithm. This means we cannot skip step 1 of Dec-Options with interpretation (b).
>
> **Interpretation (c)**  allows us to skip steps 1 and 2 of Dec-Options and it offers a baseline that is somewhat similar to the baseline Random we added to Appendix E. If $R$ is given by a fixed and repeated random set of actions, then the trajectory $\mathcal{T}$ will be given by such a random sequence. In this case, the Levin loss will be minimized by a selection of policies $R$ that choose trajectories similar to those in $\mathcal{T}$. This is similar to randomly choosing some of these policies $R$ to be used as options, as we did in Appendix E.
>
> Please let us know whether we answered your question. If not, please let us know which interpretation you had in mind: (a), (b), or (c), and we will get back to you.

---

> > ### Comment · Reviewer_voTZ · 2023-11-21
> >
> > Thank you for the explanation! I had basically the (b) variant in mind, but it's clear that the trajectories T has to be generated as well and the variant (c) now does not seem very sensible. Given the my new understanding, I am now unsure whether the baseline I previously suggested adds any value to the paper, but I will leave it to your decision.
> >
> > I am happy with the answers and changes, so I am updating the rating.

---

> > > ### Author Response · Authors · 2023-11-22
> > >
> > > Thank you for your reply and for updating the rating of our submission.

---

> > > ### Author Response · Authors · 2023-11-23
> > >
> > > We are writing to let you know that we did not forget about the paragraphs describing the baselines. We will add a detailed description of all baselines used in our experiments in the appendix of the final version of the paper.

---

### Official Review · Reviewer_i5GW · 2023-10-30

**Soundness:** 3 good
**Presentation:** 3 good
**Contribution:** 3 good
**Rating:** 6
**Confidence:** 4

**Summary:**

This paper studies the problem of extracting useful skills/options from existing neural network policies. The proposed method involves collecting a set of neural networks (that use piecewise linear activation functions) trained to perform a variety of tasks and then constructing tree structures (neural trees) representing the activation patterns of these networks. Every subtree of a neural tree can be interpreted as a separate policy giving rise to a collection of sub-policies which are used to construct options. Since this collection of options can be large and contain very specialized policies that may not be useful in general, the authors propose a greedy method for selecting a set of useful options. Experiments in grid world environments show that the extracted options are effective in learning to perform new unseen tasks.

**Strengths:**

- Inferring skills/options that can be used to perform new tasks is an important problem and the proposed approach provides an intuitive way of extracting options from pretrained neural network policies. This method leverages the compositional structure of neural networks to identify sub-policies that might be useful for performing new tasks. Based on my understanding, any such sub-policy matches the original policy on inputs triggering a specific set of activations in the network which represents a specific region of the state space where this option is used in the original policy. Therefore, this appears to be a principled way of extracting options from pretrained policies.
- The idea of directly extracting options from NN policies instead of constructing them during training appears to be novel and interesting. As far as I know, this is the first paper to propose this idea and this might lead to further research on such approaches.
- Experimental results look very promising with the proposed approach outperforming existing baselines for constructing options as well as transfer learning approaches.
- The paper is well-written and examples are provided to illustrate key concepts.

**Weaknesses:**

- The main weakness appears to be scalability of the approach w.r.t. the size of the neural network. The total number of options considered is exponential in the size of the network. The option selection method involves enumerating all candidate options which doesn’t help address the computational challenges of such an approach.
- Experiments are limited to grid world environments. Experiments in more complex environments with continuous state and action spaces would strengthen the paper significantly. Although the current experiments demonstrate the value in the proposed method for constructing options, it would be good to confirm its applicability in a wider range of scenarios.

**Questions:**

1. It appears that Levin loss corresponding to the uniform policy is equivalent to the minimum number of actions (or steps) required to explain a trajectory. Since the general Levin loss is described, are there other natural candidates (which would simply weight different actions/steps differently) besides the uniform policy to use for option selection?
1. Algorithm 1 uses $O(|\mathcal{T}|^2)$ space. It looks like the algorithm can be modified to use $O(|\mathcal{T}|)$ space (the loss after position j is independent of d). Is there a specific reason for using the version in the paper?
1. It looks like Dec-Options-Whole is doing reasonably well in most cases. Could this suggest a heuristic to reduce the computational complexity by not considering all subtrees of the neural tree? From my understanding, Dec-Options-Whole is only considering a single subtree per tree which is the whole tree, so maybe there is a middle ground between the two extremes?

---

> ### Author Response · Authors · 2023-11-21
>
> **Answer to Q1:** Wonderful question! We chose the uniform policy because it is the only safe choice to be made. Since randomly initialized neural networks represent near-uniform policies, we know that at least in the first steps of training our choice of policy will be correct. Any other choice could be harmful. For example, if we consider a policy that favors one of the options, then this could hurt exploration in a target task if the favored option is not actually helpful.
>
> **Answer to Q2:** Well spotted! Our implementation actually uses the linear version you mention. We decided to explain the quadratic because we felt it is easier to understand. In the revised version, we have added the linear version in Appendix A and added a sentence about it in the main text, after we explain the algorithm's complexity.
>
> **Answer to Q3:** Thank you for pointing this out about Dec-Options-Whole. We have a few ideas of how to move forward with the selection process with very large neural networks. The activation pattern of the network in the early tasks provides valuable information. For example, if a neuron is always active or always inactive, then we can "prune" this neuron from consideration and that would represent a three-fold reduction in the size of the space. Also, the number of activation patterns is bounded by the number of samples in the tasks, so even if we have a very large neural network, the number of patterns that matter will always be bounded by how much data we have. All this information can potentially be used to guide local search algorithms in the activation pattern space. Also, following your observation about Dec-Options-Whole, we could try to bias the selection to consider sub-policies more similar to them. We think this is an exciting research direction.

---

> > ### Comment · Reviewer_i5GW · 2023-11-23
> > **Author Response Acknowledgement**
> >
> > I thank the authors for their answers to my questions. I have read the rebuttal and skimmed through the changes in the paper. Overall, the presentation seems to have improved and the concepts are explained more clearly. Some minor concerns regarding scalability and lack of experiments on complex environments still remain. Hence, I am keeping my score.

---

> > > ### Author Response · Authors · 2023-11-23
> > >
> > > Thank you for your response. We are happy to hear you liked the changes we made to the paper.

---

### Official Review · Reviewer_j32D · 2023-10-31

**Soundness:** 4 excellent
**Presentation:** 2 fair
**Contribution:** 3 good
**Rating:** 6
**Confidence:** 4

**Summary:**

The paper proposes to decompose piece-wise linear neural networks into options. To do so the authors build on the idea of decomposing a neural network into a neural tree, a quantity closely related to oblique decision trees. As each node of the tree is a sub-policy, the authors propose to use a Levin loss to prune the important sub-policies from which options will be derived. The authors evaluate their approach on a series on grid worlds.

**Strengths:**

* The authors propose a novel way to learn options, directly from neural networks that are learning, or that have been retrained
* The overall presentation is rigorous as well as the empirical evaluation (10 seeds with 95% CI)
* The method seems to perform better compared to proposed baselines

One of the strong points of the method is the originality in the way the options are discovered. Although the empirical evaluation is limited, the method clearly has a lot of potential as stated by the authors, for example by learning options from "legacy agents". I think the community would do well in integrating such unusual ways of learning options/skills. Moreover, the empirical evaluation is rigorous and clearly shows statistical advantages.

**Weaknesses:**

* The presentation is heavy and sometimes confusing. Efforts in addressing this are done but it is not close to being enough
* The HRL baseline of Option-Critic is outdated and does not reflect progress in the field
* The qualitative experiments are very limited

The whole of section 4.1 and 4.2 would require a deep rewriting. Many references to oblique trees are made, yet most of the readers will have no idea what that is. A better visualization than Figure 1 would be needed, and it should be on the same page as the description in 4.1.  I would suggest starting with a simple case of 2 actions and leave the generalizations too much later. The notation of Z is confusing, it adds many sub- and super-scripts that are not well presented. Are the Z functions really necessary to understand the method? The whole section of 4.2 suffers from similar problems. I would strongly suggest the authors to consider the point of view of someone who knows nothing about the specifics of their method and to write these sections from that point of view. It will be most helpful to the paper.

One of the HRL baseline is Option-Critic which is an outdated algorithm that has been beaten many times. [1] recently set strong performance across a wide range of environments. To adequately understand the merits of the method such a baseline should be included.

The qualitative experiments are very limited. Much more on this aspect is needed as interpretabiltiy is one of the hallmarks of HRL. I would suggest heatmaps that show option activation or trajectories that highlight when options are activated.

**Questions:**

What would be required to scale the method to larger neural networks?

Why investigate the ComboGrid environment? What is interesting about this task?

"we consider small neural networks with one hidden layer, so we can evaluate all sub-policies of a neural policy." This should be highlighted more.


======================================================================

[1] Deep Laplacian-based Options for Temporally-Extended Exploration. Klissarov and Machado. 2023

---

> ### Author Response · Authors · 2023-11-21
>
> **Comments on Presentation**
>
> Thank you for the suggestions related to the presentation of the paper. We believe we made major improvements to the original version. Namely, the example from Figure 1 is much simpler and complete now. Instead of using a neural network with 3 neurons in the hidden layer, now we have a network with 2 neurons in the hidden layer. This allowed us to draw the entire neural tree and get rid of the formulas and of the $Z'$ notation, as you had suggested.
>
> Also aiming at improving readability, we added an example of how the Levin loss is computed with our dynamic programming procedure in Appendix A.
>
> **Suggestion of Using Klissarov and Machado's DCEO as a Baseline**
>
> Thank you for suggesting us to include the work of Klissarov and Machado (2023) as a baseline. We ran experiments with their codebase and added the results to our plots (see Figures 3 and 4 in the revised version of the paper). You will notice from the plots that we also included another recent baseline, $\epsilon$z-greedy [1].
>
> However, note that DCEO (the same goes to Option-Critic and $\epsilon$z-greedy) is not a direct competitor to the method we propose. DCEO learns options while learning a policy for a given problem. By contrast, Dec-Options are extracted from "legacy policies" and re-used in downstream problems; the options are learned even before the agent sees the target task. In some ways, Dec-Options have more information than DCEO because they use legacy policies. In other ways, DCEO has more information than Dec-Options because the options are learned for the target task, while Dec-Options relies on how well the options generalize across similar, but different tasks. Both methods make their unique and valuable contributions.
>
> **More Qualitative Results**
>
> Our way of visualizing what the options are doing is through the sequence of actions with the curly upper and lower brackets as shown at the end of Section 5.2. We also added more of such sequences in Appendix D, following your suggestion.
>
> **Question:** What would be required to scale the method to larger neural networks?
>
> **Answer:** This is a great question! We need to be able to search in the space of sub-programs and we have a few ideas of how this could be done, but we did not want to conflate the problem of searching in the space of sub-policies with the experiments that evaluate whether the sub-policies of existing networks could encode good options. We believe that the activation pattern of networks can offer valuable information on which sub-policies encode valuable information that could be transformed into options. This is a research question that needs to be addressed.
>
> **Question:** Why investigate the ComboGrid environment? What is interesting about this task?
>
> **Answer:** The Combo domain is a hard exploration problem that allows us to easily understand the sub-policies extracted from the neural networks. Moreover, the domain has clear dynamics where options could be helpful. Human designers would possibly design one option for each movement the agent can perform (up, down, left, and right). We wanted to see whether Dec-Options would be able to find something similar without having any prior information of the problem. We were pleased to see that our method discovered options similar to what we would have designed.
>
> **Comment:** "we consider small neural networks with one hidden layer, so we can evaluate all sub-policies of a neural policy." This should be highlighted more.
>
> **Answer:** Good point! We added a sentence in the last paragraph of the Introduction highlighting this methodological choice.
>
> **Reference**
>
> [1] Dabney, W., Ostrovski, G., and Barreto, A. Temporally-extended ϵ-greedy exploration. In International Conference on Learning Representations, 2021.

---

> > ### Comment · Reviewer_j32D · 2023-11-22
> >
> > Dear Authors,
> >
> > thank you for your detailed rebuttal and for adjusting the paper with clarifications and a new baseline.
> >
> > The papers has cleared improved in terms of presentation and claims. I also appreciate the visualizations that help understand the method. I would still have some concerns in terms of scalability and fair comparisons to baselines, but the novelty and originality of the work outweighs these concerns. I am hopeful that this paper could propose a different line of work on learning temporal abstractions. I am raising my score.

---

> > > ### Author Response · Authors · 2023-11-23
> > >
> > > Thank you for your response and for being supportive of our line of work.

---

### Author Response · Authors · 2023-11-21
**All Reviewers**

We sincerely thank all reviewers for their contributions. We believe our paper has improved once we took the reviewers suggestions into account. Here are the main changes we made:

1. We changed the example in Figure 1. Following the suggestions of the reviewers j32D and voTZ, we now present a smaller neural network for which we can show the entire neural tree. This also allowed us to get rid of the $Z'$ notation and of the equations we had in Figure 1. We only kept the notation of neural networks, which is standard.
2. Following the suggestion of i5GW, we added a version of the compute-loss algorithm to Appendix A that runs in linear memory on the length of the trajectories. We also added an example of how the loss is computed, following j32D's suggestion of providing a more detailed explanation of the method.
3. We added experimental results of the method by Klissarov and Machado (2023), as j32D had suggested (see Figures 3 and 4 of the revised paper).
4. We added experimental results of the method by Dabney et al. (2021), as another recent baseline, following the suggestion of j32D (see Figures 3 and 4 of the revised paper).
5. We added the baseline that ignores steps 1 and 2, as voTZ suggested (see 'Random' in the plots of Figure 10 of the revised paper).
6. We added more sequences of option activations in Appendix D, following the suggestion of j32D to include more qualitative results. These sequences are somewhat similar to the heatmaps of option activations, as they show what each option is doing.

We have uploaded a new version of the paper where the changes are highlighted in blue.

Please let us know if there is anything else you would like to see changed in the paper or if you need any additional information regarding the questions raised in the reviews.

---

> ### Author Response · Authors · 2023-11-21
> **Are We Restricted to Small Networks?**
>
> **Short answer:** No.
>
> **Long answer:** We performed all our experiments on small neural networks so we would be able to evaluate all sub-policies of a given network. This way, we would not conflate a possible lack of helpful sub-policies with our potential inability to search for helpful options.
>
> However, this does not mean that our method is restricted to small neural networks. Reviewer i5GW correctly pointed out that the variant Dec-Options-Whole, which only considers one sub-tree of the neural tree and does not pose constraints on the size of the neural network, already provides strong results.
>
> Moreover, now that we know that it is possible to extract helpful policies with network decomposition, future research will investigate search algorithms for the space of sub-policies.

---

### Author Response · Authors · 2023-11-23
**Thank you!**

We would like to thank all reviewers again for reading our paper and providing thoughtful feedback on our work. We also would like to thank them for engaging with us in the discussion period. We believe our paper has improved substantially thanks to their feedback.

---

### Meta-Review · Area_Chair_jua7 · 2023-12-06

**Metareview:**

The paper proposes a method to extract options from a learned neural network . They provide an algorithm that distills the neural network into a neural tree, and then prune the tree based on the Levin loss to choose the most promising options. The experiments show the success of the proposed approach on some grid-based environments. The authors have incorporated many of the reviewers suggestions, but scalability of the approach remains somewhat understudied.

**Justification For Why Not Higher Score:**

It is unclear how well the method would scale. Experiments are only provided on grid-based environments.

**Justification For Why Not Lower Score:**

The proposed technique is an interesting way to directly extract options from a trained neural policy. This might have applications to explainability and interpretability research as well.

---

### Decision · Program_Chairs · 2024-01-16

Accept (poster)